# Position: Generative AI Regulation Can Learn from Social Media Regulation

Ruth E. Appel [1]

## Abstract

There is strong agreement that generative AI should be regulated, but strong disagreement on how to approach regulation. While some argue that AI regulation should mostly rely on extensions of existing laws, others argue that entirely new laws and regulations are needed to ensure that generative AI benefits society. In this position paper, we argue that the debates on generative AI regulation can be informed by evidence on social media regulation. For example, AI companies have faced allegations of political bias which resemble the allegations social media companies have faced. First, we compare and contrast the affordances of generative AI and social media to highlight their similarities and differences. Then, we discuss four specific policy recommendations based on the evolution of social media and their regulation: (1) counter bias and perceptions thereof (e.g., via transparency, oversight boards, researcher access, democratic input), (2) address specific regulatory concerns (e.g., youth wellbeing, election integrity) and invest in trust and safety, (3) promote computational social science research, and (4) take on a more global perspective. Applying lessons learnt from social media regulation to generative AI regulation can save effort and time, and prevent avoidable mistakes.

## 1. Introduction

When Google's generative AI model Gemini produced images of racially diverse Nazis in early 2024, it led to a public outcry and allegations of anti-conservative bias (Robertson, 2024). Almost a decade earlier, the first allegations of anti-conservative bias were made against social media platforms like Facebook (Barrett & Sims, 2021), and they have persisted e.g. during Senate hearings (Romm, 2019)

and when President Trump was banned from Twitter (now X) and Facebook (Barrett & Sims, 2021). This shows that the content moderation challenges that emerging technologies face are not entirely new. Media scholars have called attention to the fact that new technologies often elicit similar questions and concerns as their predecessors (Wartella & Reeves, 1985). Generative AI is the latest technology to garner widespread attention and raise societal and regulatory concerns, but so have social media and other technologies before it.

In this paper, we argue that **generative AI regulation can learn from social media regulation**, which has evolved over the past two decades. While there is strong agreement that generative AI should be regulated—evidenced by the large number of recent regulatory efforts across countries and stakeholders (Zaidan & Ibrahim, 2024)—, there is strong disagreement on how to approach regulation. Some argue that AI regulation should mostly rely on extensions of existing laws (Huttenlocher et al., 2023), while others argue that entirely new laws and regulations are needed and have proposed laws and regulations such as the EU AI Act (European Parliament and Council, 2024), White House Executive Orders on AI (Executive Office of the President, 2023; 2025), or California's vetoed AI Safety Bill SB 1047 (Wiener et al., 2024). Analyzing the evolution of social media regulation can provide insights into which approaches to regulation are promising when it comes to generative AI, which in turn can prevent avoidable mistakes, and save effort and time.

Learning from social media regulation is imperative because AI regulation is urgent. Misused or misaligned generative AI can cause severe harms (Weidinger et al., 2021; Marchal et al., 2024; MITRE Corporation, 2021), and the risks become even greater as generative AI advances (Hendrycks et al., 2025). Yet, despite a patchwork of emerging global regulations, effective global regulation of generative AI is lacking (Alanoca et al., 2025).

Specific learnings for generative AI regulation based on the evolution of social media regulation include investments in (1) efforts to counter bias and perceptions thereof (e.g., via transparency, researcher access, oversight boards, democratic input), (2) specific areas of regulatory concern and trust and safety, (3) computational social science research,

[1]Stanford University, Stanford, CA, United States. Correspondence to: Ruth E. Appel <rappel@cs.stanford.edu>.

*Proceedings of the $42^{nd}$ International Conference on Machine Learning*, Vancouver, Canada. PMLR 267, 2025. Copyright 2025 by the author(s).

and (4) a more global perspective (see Section 3).

The focus of this paper is on content moderation, i.e. how to design and regulate the content generated by generative AI models or shown on social media platforms, because social media regulation holds particularly relevant insights in this area. Further, the paper focuses on regulation in a broad sense, which can include self-regulation of industry players and formal laws such as the EU AI Act (European Parliament and Council, 2024) or White House Executive Orders on AI (Executive Office of the President, 2023; 2025).

First, we compare and contrast the affordances of generative AI and social media to highlight their similarities and differences. Second, we provide policy recommendations based on the evolution of social media and their regulation. Third, we discuss implementation challenges such as political polarization and the pace of technological change. Fourth, we engage with alternative views before we conclude.

## 2. Affordances of Generative AI and Social Media

To shed light on the similarities and differences between specific media, we can analyze their affordances. For the purposes of this paper, we define affordances as the features that characterize a medium in its relationship to its users (for a detailed discussion of different definitions and the evolution of the term affordances, see McGrenere & Ho, 2000; Ronzhyn et al., 2023). Both generative AI, e.g. in the form of a chatbot like OpenAI's ChatGPT or Anthropic's Claude, and social media, e.g. in the form of Meta's Facebook or X (formerly Twitter), can be considered media that allow to create and distribute content and are shaped by specific features. The features discussed here pertain to a medium in general, but may not apply to every instance, that is, a specific generative AI application or social media platform may differ from the norm in terms of its affordances.

Based on an analysis of commonly used generative AI applications (e.g., ChatGPT and Claude) as well as social media applications (e.g., Facebook and X), we identified key features that generative AI and social media share or that differentiate them. The analysis of features is grounded in work by Clark (1996), who discusses several features of media that fall into three categories: medium, control, and immediacy. Since Clark (1996)'s features focus on the affordances of face-to-face communication,[1] we added new features and removed features that are less relevant to the

---

[1] There are contextual differences between face-to-face communication on the one hand and generative AI and social media on the other, such as where and why they may be used. This paper focuses on the comparison of generative AI and social media, and therefore focuses on features in Clark (1996)'s model that are pertinent to generative AI and social media, but not the comparison to other media.

comparison of generative AI and social media. We also added the feature of interactivity discussed by Rafaeli & Sudweeks (1997). We will point out each feature that is adapted from Clark (1996) or Rafaeli & Sudweeks (1997).

We will first address why social media is comparable to generative AI in key aspects that have implications for technology regulation. Then, we will engage with differences in the affordances of social media and generative AI to show that the analogy is useful, but imperfect.

### 2.1. Generative AI and Social Media Are Comparable in Key Aspects

The analogy between generative AI and social media is valuable because both media share key features. Importantly, the shared affordances of generative AI and social media imply that both of these media necessarily moderate content and thus face complex content moderation challenges and public scrutiny.

Table 1 shows key similarities between generative AI and social media when it comes to the features of each *medium*. Both generative AI and social media allow for *spatial separation*, that is, the conversation partners usually generate content in different physical spaces—e.g., in a home office and at a data center for generative AI—and are not copresent (copresence is one of the features described in Clark (1996)). Both media feature **interactivity** and respond interactively to user input, which makes them engaging (Rafaeli & Sudweeks, 1997) (interactivity is defined and discussed in Rafaeli & Sudweeks (1997)). Both generative AI and social media are *recording* user data (the recording feature is adapted from Clark (1996)'s recordlessness feature). Both media can learn about a user's context and their preferences over time for output *personalization*, e.g. by updating the chatbot's memory or personalizing a recommendation algorithm. Further, both generative AI and social media can feature *general content*, i.e. content on all kinds of domains (e.g., hobbies, jobs, politics). Both are powered by *artificial intelligence* (AI), that is, they rely on learning patterns from data to perform well on tasks such as generating text or recommending content, although generative AI relies on more recent deep learning models while social media tends to rely on traditional machine learning approaches such as recommender systems. Both media also feature *abstraction*, that is, they hide the complex technical implementation details from the user behind a simple user interface. Further, generative AI and social media algorithms tend to be *blackbox*, that is, algorithmic decisions are intransparent—almost always for users, but often also for experts because mechanistic interpretability (Bereska & Gavves, 2024) that can explain why a deep learning model made a certain decision is in its infancy.

With regards to **control** features (Clark, 1996), both gen-

*Table 1.* **Comparison of affordances of generative AI and social media**

| Feature | Definition | Generative AI | Social Media |
|---|---|---|---|
| *Medium* | | | |
| Spatial separation | Content is generated in different locations | **Yes** | **Yes** |
| Direct connection | Medium is conversation partner | Yes | No |
| User connections | Medium connects user to other users | No | Yes |
| Interactivity | Medium responds interactively to user input | **Yes** | **Yes** |
| Dialogue-by-default | Actions occur in a dialogue | Yes | No |
| Recording | User actions are recorded | **Yes** | **Yes** |
| Personalization | User context and preferences are learnt over time | **Yes** | **Yes** |
| Single output | Medium presents usually just a single output | Yes | No |
| Infinite content | Content is served infinitely | No | Yes |
| General content | Content can pertain to any domain | **Yes** | **Yes** |
| General purpose | Medium serves many functions | Yes | No |
| Use of AI | Medium learns patterns from data | **Yes** | **Yes** |
| Abstraction | Medium hides its complexity | **Yes** | **Yes** |
| Black-box | How algorithmic decisions are made is intransparent | **Yes** | **Yes** |
| *Control* | | | |
| Content moderation | Content is moderated at all | **Yes** | **Yes** |
| Invisible content moderation | Most content moderation is not visible to the user | Yes | No |
| Content moderation pre-generation | Content is moderated before it is received by the user | Yes | No |
| Self-determination | User can decide themselves how to act | **Yes** | **Yes** |
| Self-expression | User can express themselves | **Yes** | **Yes** |
| Simultaneity | User can receive and produce content concurrently | No | Yes |
| *Immediacy* | | | |
| Instantaneity | Actions are perceived almost immediately | **Yes** | **Yes** |
| Evanescence | Medium quickly recedes to the background | **Yes** | **Yes** |

*Note*: The features spatial separation, recording, self-determination, self-expression, simultaneity, instantaneity, and evanescence, as well as the categories medium, control and immediacy are based on Clark (1996). The feature interactivity is based on Rafaeli & Sudweeks (1997). Instances where features of generative AI are similar to features of social media are highlighted in bold.

erative AI and social media feature *content moderation*, that is, the medium shapes what content is allowed to appear. Both media also meet Clark (1996)'s criteria for *self-determination*, i.e. a user's ability to decide themselves how to act, and *self-expression*, i.e. a user's ability to express themselves on a medium.

With regards to **immediacy** (Clark, 1996), both generative AI and social media share *instantaneity* (Clark, 1996), i.e. that actions are perceived almost immediately, and *evanescence* (Clark, 1996), i.e. that the medium recedes to the background quickly once it is not actively used anymore.

Beyond features, the evolution of generative AI is similar to the evolution of social media in that both are characterized by limited, lagging regulation and large inflows of funding for technology entrepreneurship in this space (Stern, 2023).

### 2.2. Generative AI and Social Media Are Not Perfectly Comparable

While the shared affordances highlight the value of comparing generative AI to social media, we acknowledge that the analogy is imperfect. By definition, an analogy is not a perfect match. As Jacob Stern put it: "[T]his is just the nature of analogies: They are illuminating but incomplete" (Stern, 2023).

Table 1 reveals differences in affordances between generative AI and social media. With regards to features of the **medium**, generative AI and social media show some variation. While generative AI such as ChatGPT constitutes a conversation partner that is in *direct connection* with the user, social media foster *user connections*—connections between users. These differences in connection also imply that generative AI tends to be more private by default, since conversations are rarely shared with other users. Whereas generative AI interacts in a *dialogue-by-default* manner with the user, social media is merely mediating between the user and their human conversation partners (e.g., when a social media algorithm displays one user's post on another user's feed) and tend to involve a sequence of one-off actions. While generative AI tends to respond to prompts, usually with a *single output* instead of multiple outputs, and does not continue to serve content unless the user requests it, social media often feature *infinite content* via mechanisms such as infinite scroll (Sharma & Murano, 2020) or autoplay (Lukoff et al., 2021), which serve content as long as the user is on the platform and encourage passive rather than active consumption. The purpose of social media tends to be focused on social communication, while generative AI is considered a *general purpose* technology that could serve various functions, including as a text writer or reviewer, a

calculator, a programmer and much more.

With regards to **control** features, a feature Clark (1996) proposed is *simultaneity*, which is the user's ability to receive and produce content concurrently. Simultaneity is given for social media—e.g., one user might send a message at the same time as another user is sending them a message—, but not for generative AI, which operates in a sequential dialogue of user input and model output. Important differences between generative AI and social media are related to content moderation: Even though both generative AI and social media feature content moderation, content moderation in generative AI tends to use *invisible content moderation* more than social media. Social media platforms may occasionally take hardly visible actions such as downranking posts, but many social media content moderation actions such as removal of a post or user are clearly visible. Generative AI models, on the other hand, are built and fine-tuned to moderate content in a certain way (e.g., to avoid providing dangerous information), without the user necessarily becoming aware of the moderation. Generative AI content moderation may be invisible to the user because the model will usually respond, and not necessarily provide a reason if it refuses to respond to a prompt directly, which makes moderation less obvious than a missing response or a refused response citing the reason for refusal. Relatedly, generative AI models tend to moderate *before* the content is shown to the user, e.g. by refusing to reply to a prompt, while social media content moderation tends to occur only *after* content made it onto a platform, e.g. when a post was reported as harmful misinformation.

Beyond specific features of generative AI and social media, there are differences in their context and potential consequences. In terms of business model, most social media companies rely on revenue from advertisements (Center for Humane Technology, 2021), while prominent generative AI companies have so far leaned towards freemium (Kumar, 2014) subscription models. While the potential harm of social media to democracy and society has been an important focus of scholarly and public attention (Persily & Tucker, 2020), some argue that the destructive potential of AI may be at another level since it may present a larger threat (Bostrom, 2013) or stronger geopolitical advantage (Stern, 2023). Generative AI and social media differ also in the level of uncertainty they bring. For example, auditing and discovering vulnerabilities in systems that are probabilistic (Cattell et al., 2024), like generative AI models, implies new complexities that traditional, deterministic social media algorithms do not entail. Finally, generative AI and social media may differ in areas that have so far remained legally uncertain, such as questions of liability (e.g., for harms resulting from media use) and copyright. This means the learnings for generative AI regulation should be based on, and not go beyond key shared features.

## 3. Learnings from Social Media Regulation for Generative AI Regulation

As the review of the affordances has shown, generative AI and social media share important features, including the use of AI and content moderation. Although generative AI and social media differ on some dimensions, these differences are mostly differences in degree, and not differences in kind when it comes to regulation. Thus, lessons learnt from social media regulation are relevant to generative AI regulation. This paper provides four policy recommendations for generative AI regulation based on the evolution of social media regulation: (1) counter bias and perceptions thereof (e.g., via transparency, oversight boards, researcher access, democratic input), (2) address specific regulatory concerns (e.g., youth wellbeing, election integrity) and invest in trust and safety, (3) promote computational social science research, and (4) take on a more global perspective. Figure 1 provides an overview of these recommendations.

### 3.1. Counter Bias and Perceptions Thereof

Given that both generative AI and social media share key features—use of content moderation, use of AI, blackbox nature, abstraction of the complexity of algorithmic decision-making such that much of the decision-making is intransparent—, it is no surprise that both generative AI companies and social media companies have faced allegations of bias, including allegations of anti-conservative political bias (Robertson, 2024; Barrett & Sims, 2021). While there is no evidence of anti-conservative bias for social media (Barrett & Sims, 2021), multiple studies have shown political bias in generative AI. For example, compared to representative opinion polls, large language models were found to output biased opinions (Durmus et al., 2023; Santurkar et al., 2023), and multiple studies showed left-leaning bias in generative AI models (Rozado, 2023; Röttger et al., 2024).

Generative AI models have also been shown to exhibit other forms of bias, such as anti-Muslim bias (Abid et al., 2021), bias towards Western culture (Naous et al., 2023), and stereotypical depictions of race, gender, age, nationality, and socioeconomic status (Nangia et al., 2020). Similarly, generative AI models tend to show social identity biases similar to humans (Hu et al., 2024).

Addressing such biases is as important as it is challenging. It is important to address biases because biases can harm and manipulate users. For example, political bias in generative AI models can influence users' opinions (Bai et al., 2023; Jakesch et al., 2023; Matz et al., 2024; Williams-Ceci et al., 2024; Potter et al., 2024; Anthropic, 2024) and decisions (Fisher et al., 2024). Biases may also lead to lower-quality output, entrench historical biases and stereotypes, and undermine trust. It is challenging to address biases because

| Policy Recommendation | Key Strategies | Proposed Generative AI Regulation Measures | Social Media Precedents |
|---|---|---|---|
| **Counter bias or perceptions thereof** | Transparency, oversight, researcher access, democratic input, personalization | Researcher API access, transparency requirements, decentralization | Meta Transparency Center, TikTok Research API, Mastodon's decentralized content moderation |
| **Address specific regulatory concerns and invest in trust and safety** | Investment in promoting youth wellbeing, election integrity, and misinformation prevention | Dedicated trust and safety teams, deceptive campaign monitoring | Trust and safety teams at social media companies such as Google and Meta |
| **Promote computational social science research** | Multidisciplinary study of platform impact, evaluation of interventions | AI user experience research, interdisciplinary hiring, rigorous impact evaluation | Facebook and Instagram Election Study, multidisciplinary in-house research teams |
| **Take a more global perspective** | Local expertise, international hiring, multilingual content moderation | Ensuring safety and performance in diverse contexts, regionally adapted safety policies | Regional regulatory adaptation, Christchurch Call |

*Figure 1.* **Policy recommendation overview.** Overview of the lessons generative AI regulation can learn from social media regulation.

they are challenging to measure accurately. For example, bias evaluations may be sensitive to the specific prompt design (Röttger et al., 2024) and order effects (Dominguez-Olmedo et al., 2024). Further, it is not clear where exactly biases stem from. Biases can arise at different points in the development and deployment of generative AI, including training and data curation, fine-tuning, evaluation and feedback, real-time moderation, customization and control of models (Suresh & Guttag, 2021; Ferrara, 2023).

Social media companies have taken different approaches to address biases or perceptions thereof that mainly focus on transparency about algorithms and decision-making, gathering input from users and learning from case studies, and increasing user choice.

### 3.1.1. INCREASE TRANSPARENCY AND RESEARCHER ACCESS

The shared features content moderation, use of AI, black-box and abstraction give rise to transparency challenges for social media and generative AI. Generative AI transparency is lacking as shown by the Foundation Model Transparency Index (Bommasani et al., 2023a; 2024). Social media companies have pursued multiple different approaches to increase transparency and generative AI can learn from this playbook. For example, Facebook's parent company Meta introduced features such as "Why am I seeing this ad?" that allowed users to understand why they were served certain ad content (Thulasi, 2019), created blog posts and a Transparency Center providing some information on the role of AI and other factors in content recommendation (Clegg, 2023; Meta, 2024a), and established an independent oversight board of experts that adjudicates particularly contentious content moderation decisions (Meta, 2024b). These initiatives do not come without problems. In response to the launch of Facebook's oversight board, "The Real Facebook Oversight Board" was created, which brought experts together to argue for more independence, transparency

and regulation (The Real Facebook Oversight Board, 2022). Company policies are also not guaranteed to be permanent. In January 2025, Meta starkly shifted its content moderation policy, limiting its efforts to reduce misinformation and harmful speech and ending a fact-checking program that had provided some transparency about the content circulating on the platform (Isaac & Schleifer, 2025; Iyer, 2025).

An important aspect of transparency is allowing for third-party evaluations. Efforts to create research platforms or APIs accessible to researchers, such as the Meta Researcher Platform (Li et al., 2022) and the TikTok Research API (TikTok, 2025), or to design academic-industry collaboration such as the Facebook and Instagram Election Study (Clegg & Nayak, 2020) are helpful, but imperfect (Wagner, 2023). The Coalition for Independent Technology Research was founded after researchers at different institutions faced difficulty maintaining or gaining access to social media data for research purposes (Coalition for Independent Technology Research, 2022). Importantly, we can learn from these shortcomings. Researcher access programs to evaluate technology should be characterized by sufficient resources (including staffing, infrastructure, and funding), incentives that are compatible with academic research (e.g., data retention policies, persistent API access and publication permission for researchers), sound knowledge sharing processes between internal and external researchers to help understand data availability and analysis feasibility, helpful documentation, privacy preserving measures (e.g., aggregation of user data) and timeliness in terms of data access, publication review and addressing issues that researchers discovered. To protect researchers involved, researchers have called for "safe harbors," that is, legal protection for researchers pursuing legitimate research purposes, initially for social media (Abdo et al., 2022) and more recently for generative AI (Longpre et al., 2024). Additional proposals to facilitate external generative AI research include data donations (Sanderson, 2024).

Regulations like the Digital Services Act prescribe transparency by requiring audits of social media companies (European Commission, 2023), and similar auditing efforts are imaginable for generative AI. In fact, some scholars suggest to extend and adapt DSA rules for social media platforms to generative AI (Hacker et al., 2023).

While the specific implementation of these transparency efforts may be contentious and requires nuance, there is a broader lesson: Generative AI regulation can incentivize measures for increasing transparency, such as short and accessible explanations of the technology, independent oversight mechanisms, researcher access and mandatory audits.

### 3.1.2. GATHER DEMOCRATIC INPUT TO INFORM TECHNOLOGY

Generative AI and social media share features that make them complex, including that the content they feature can pertain to a variety of domains, that there is potential for personalization, and that content could be moderated in various different ways. One approach to determine what a good content moderation system may look like is to gather input directly from users to inform design choices. Different initiatives have been launched over the past few years to gather input from users and enable democratic decisions about the nature of regulation and content moderation, with users deliberating issues ranging from cyberbullying on social platforms to the rules and constitutions that inform generative AI models (Wetherall-Grujić, 2023). These initiatives have their roots in the idea of deliberative democracy (Eagan, 2016). Social media also offers case studies of networks where content moderation seems to be broadly accepted and deliver productive results, such as in the case of the deliberation platform vTaiwan (Miller, 2019) or a neighborhood-focused social network (Oremus, 2024). Finally, researchers have studied how to embed important societal values into social media AI (Bernstein et al., 2023), which could inform how such values can be embedded into generative AI.

### 3.1.3. PROMOTE USER CHOICE

Another option to empower users to make choices in the face of features such as content moderation and the varied nature of content is to enable users to set up rules for a subset of the system. The social media platform Mastodon is a prominent example in terms of increasing user choice in such a way. Mastodon is built on the idea that different communities can create their own servers and set and enforce their own content moderation rules (Mastodon, 2024). This highlights that the feature of personalization may be a potential route for resolving content moderation dilemmas. Content moderation questions with regards to generative AI and social media are similar and it is not clear what opinion represen-

tation should be the default, but increased personalization of models may be an answer (Redpoint, 2020).

### 3.2. Address Specific Regulatory Concerns and Invest in Trust and Safety

The feature of content moderation that generative AI and social media share comes with challenges such as preventing the spread of harmful misinformation and protecting user wellbeing. Social media companies have invested in teams that address these specific regulatory concerns. Examples include teams at companies like Google, Meta and Microsoft working on youth wellbeing and mental health in general, election integrity, preventing spam, preventing the spread of child sexual abuse material, preventing harmful misinformation, detecting deceptive campaigns, and ensuring trust in the platform and safety of its users in general.

Generative AI chatbot performance has already been rated with regards to certain principles that apply just as much to social media. Common Sense Media published rankings of different generative AI models with respect to the following principles: put people first, prioritize fairness, be trustworthy, keep kids and teens safe, be effective, help people connect, use data responsibly, and be transparent and accountable (Common Sense Media, 2024).

Yet, generative AI companies do not have teams at the same scale as social media companies to address these issues. Generative AI companies are much smaller and younger than some of the social media giants, thus it is not surprising that they do not have as much dedicated staff to work on these issues. Going forward, however, adding diverse staff beyond engineers that can bring in expertise to address issues such as user mental health or combating misinformation is important to address the variety of risks and harms that generative AI models pose (for taxonomies of risks and harms related to generative AI, see Weidinger et al., 2021; Marchal et al., 2024; Gabriel et al., 2024; MITRE Corporation, 2021). Investment in trust and safety teams seems particularly crucial, and it is encouraging to see that companies like OpenAI and Anthropic are investing in this area, with OpenAI publishing the first-ever report on the activity of deceptive campaigns on generative AI platforms in May 2024 (Nimmo, 2024).

The policies social media companies have put in place to decide how and when to moderate individual users, and the best practices they have developed to uncover abuse such as deceptive campaigns that try to interfere with elections or spam users, could inform the approaches generative AI companies take. This includes developing a repertoire of content moderation approaches, which could include bans, but also more cautious interventions such as warnings and strikes for misbehavior, putting more guardrails in place or throttling usage for users that try to abuse generative AI

models. Social media companies also gained experience in involving the user community in content moderation decisions (e.g., in the case of BirdWatch (Wojcik et al., 2022)) and how to collaborate across platforms, and generative AI companies could consider how these approaches could be adapted to their platforms.

Importantly, implementation of trust and safety measures for generative AI does not have to start from scratch. Open-source, collaborative tools like the Robust Open Online Safety Tools (ROOST, 2025) are a concrete example of collaboration across platforms and enable access to trust and safety resources even for companies with limited resources.

### 3.3. Promote Computational Social Science Research

Both generative AI and social media allow users to express themselves and allow for a connection, be it to other users or to an AI with a vast pool of knowledge. How these media interact with users is a key part of what makes them so influential. They are neither purely technical, nor purely social systems. This suggests that multidisciplinary study—computational social science—is needed to understand, evaluate and shape these systems (Gillespie et al., 2024).

In fact, the recommendations above, whether regarding measures to reduce bias or to enhance user wellbeing, all require computational social science research to test their effectiveness. Social media companies have hired researchers from many disciplines, including computer science, psychology, political science, communication, law and others, to better understand how their platforms impact society, and how certain interventions influence society and their revenue.

Rigorous computational social science evaluations, whether conducted in-house or via external researchers with platform access, are key to ensuring that technologies such as generative AI and social media meet their goal of being helpful and not harmful to society. Further investment in research is needed because generative AI has features that differ from previous technologies, so its impact and user preferences (e.g., with regards to privacy, personalization or content moderation) are not clear. Even the impact of previous technologies such as social media has not yet been comprehensively evaluated and needs further investment. Rigorous research can inform platform and public policy when it comes to regulation, and it can enhance user trust.

This implies the need to invest in diverse research teams that understand the interaction of humans and technology and can evaluate the societal implications technology. While AI company recruiting often focuses heavily on engineers, and some companies are more concerned with extreme risks in the more distant future, social media companies have shown the value of creating multidisciplinary teams to address current risks such as biases. Multidisciplinary teams allow companies to test different product features and interventions effectively, e.g. to reduce spam or misinformation spread. Guidance on building effective red teams for generative AI models also highlights the importance of diverse teams (Ofcom, 2024; Metcalf & Singh, 2024; Ahmad et al., 2024; Oremus, 2023). Computational social scientists from any background, data scientists and user experience researchers would be especially helpful to address questions at the intersection of humans and technology, such as which emotional bonds may be formed between humans and AI, and what type of personalization should be implemented.

While content moderation on social media is far from a resolved issue, there is a large and growing body of academic literature addressing user preferences and content moderation approaches (e.g., Persily & Tucker, 2020; Appel et al., 2023; Kozyreva et al., 2024), which could inform content moderation for generative AI.

### 3.4. Take on a More Global Perspective

As the features spatial separation, general content, and use of AI imply, both generative AI and social media can be used in a variety of contexts. Generative AI companies have grown rapidly and are serving users around the world, similar to social media companies. However, compared to social media companies, many generative AI companies are more heavily focused on the US, likely due to their headquarter location (with exceptions such as Google DeepMind in the UK, Mistral AI in France, and DeepSeek in China). To address problems like biases, it is crucial that even small companies take on a global perspective and embrace local expertise in multiple countries. The reasoning mirrors that for the benefits of diversity in AI red teaming (Ofcom, 2024; Metcalf & Singh, 2024; Oremus, 2023), i.e. that broader representation allows for a better understanding of user preferences and the harms that a technology may pose. Taking on a more global perspective could take the form of establishing local offices and a focus on hiring internationally. The stakes are high. If companies fail to invest in taking user preferences and risk factors outside of the US seriously, the technology may serve large numbers of users worse (e.g., due to under-investment in non-English language content generation), contain undiscovered harms (Metcalf & Singh, 2024; Oremus, 2023), and could even result in catastrophes such as promoting violence in conflict regions (Amnesty International, 2022). Given the increasing amount of national and local regulations on generative AI, global expertise is also important to keep up with local laws.

For effective regulation, local expertise needs to be integrated into a global perspective. For example, the former Prime Minister of New Zealand suggested that a model for governing AI could follow the Christchurch Call, which is a multinational, multi-stakeholder effort bringing together

governments, tech companies and civil society to eliminate violent extremist and terrorist content online (Ardern, 2023).

## 4. Discussion

Applying learnings from social media regulation to generative AI regulation is challenging, in particular in light of political polarization, the rapid pace of technological development, the need to take different stakeholder characteristics into account, the patchwork of emerging AI regulations around the globe, and the need for effective implementation.

Policy is shaped by the contemporary political context. Thus, it is important to acknowledge that high levels of political polarization (documented e.g. in Finkel et al., 2020; Ruggeri et al., 2021) make it more difficult to develop policy that enjoys broad support (Druckman et al., 2021). When it comes to AI, public opinion and trust in governments and AI companies vary greatly based on people's characteristics, such as partisanship and nationality (Ipsos, 2024; Mcclain et al., 2025; Dreksler et al., 2025). Generative AI regulation has to be designed with this political context in mind. While polarization is a deeply-rooted issue that is difficult to address, our hope is that the recommendations on countering bias and perceptions thereof, increasing transparency and researcher access, and gathering democratic input to inform technology design may help depolarize the debate.

Another challenge for generative AI regulation is the pace of technology development. The capability leaps and proliferation of new models suggest that generative AI development outpaces social media development. This points to the need for more flexible policy, which is designed with foresight and is adaptable to future changes in the technology stack.

Further, effective regulation needs to take different stakeholder characteristics into account. For example, AI developers differ in their size, products, popularity, and resources. This implies that different stakeholders face different challenges in implementing regulations and managing compliance burden. Thus, regulations should be flexible. For example, the EU AI Act (European Parliament and Council, 2024) has different requirements depending on company size, user base size, compute used to create AI systems and whether AI systems are open source.

As argued in the previous section, generative AI regulation needs a global perspective. Yet, it has to account for a patchwork of different AI regulations around the globe (Alanoca et al., 2025). Some of these regulations are more compatible with the recommendations in this paper than others, which makes the implementation of these recommendations more feasible in some jurisdictions. The EU AI Act (European Parliament and Council, 2024) is one of the most comprehensive AI regulations around the globe. It regulates based on the level of risk that an AI system poses and takes other factors such as openness of the technology into account. The EU AI Act aligns with several of the recommendations above, including transparency requirements, researcher access requirements, and a focus on specific regulatory concerns and risk areas. Work on the EU AI Act began before generative AI was widely adopted, but expanded its scope to include generative AI as it emerged. Thus, the EU AI Act addresses both traditional AI, such as social media, and generative AI, which may allow to integrate learnings from social media. Other regulation that was originally designed with technologies such as social media in mind is the EU Digital Services Act (DSA) (European Parliament and Council, 2022). This regulation also features transparency requirements, researcher access requirements, and specific guidelines when it comes to regulatory concerns such as election interference. Some argue that DSA rules could be adapted for generative AI platforms (Hacker et al., 2023). Another pertinent and controversial regulation that shaped social media is Section 230 of the Communications Decency Act (U.S. Congress, 1996). Section 230 holds that interactive computer service providers, including social media companies, are not considered publishers or speakers when they provide information that was provided to them by other users, which greatly limits platforms' liability for problematic content shared by users. This relates to generative AI regulation because AI developers face similar liability questions: While AI developers provide information that their technology generated and they could be considered publishers or creators, their content generation is based on training on others' speech and they could be seen as intermediaries. Section 230 has been the subject of intense legal and political debate, with uncertain outcomes. The debates about Section 230 demonstrate that definitions and liability determinations can evolve and can shape platforms' business models and content moderation strategies. Learning from existing regulation before adding new ones can help address the lack of specificity of the current environment, and prevent further fragmentation.

A final challenge for effective regulation is effective implementation. Regulation should be concrete enough to allow for smooth implementation by stakeholders with different characteristics—regulations and standards that are too vague impose a significant burden on stakeholders and are difficult to monitor (Pouget & Zuhdi, 2024). More generally, it is crucial to track readiness and compliance (see e.g. Scott, 2024; Bommasani et al., 2023b) and ensure proper evaluation and incentives.

## 5. Alternative Views

In this paper, we argued that generative AI regulation can learn from social media regulation. However, there are valid counterarguments related to the imperfect analogy between

generative AI and social media, the fact that social media regulation has not been a model example of technology regulation, and that thinking about regulation from first principles may be desirable.

First, as discussed in detail in Section 2.2, there are important differences in the affordances of social media and generative AI, including whether the medium acts as conversation partner, how visible content moderation is, and how many functions the medium serves. For example, generative AI and social media differ in that only generative AI chatbots tend to be a direct conversation partner for humans, in contrast to traditional social media platforms where people post to interact with other people. This could have implications for the kind of relationships that people form with the technology, which in turn could affect the need for regulation. While this concern is valid, the recommendations in this paper build on the affordances that generative AI and social media share and do not go beyond those. For differences in affordances, other comparisons could be insightful (for a review of different AI metaphors, see Maas, 2023). For example, learning from regulatory authorities such as the FDA could help inform AI governance (Raji et al., 2022; AI Now Institute, 2024).

Second, as described in earlier parts such as Section 3.1.1, social media regulation has not been a model example of technology regulation. Research projects like the Facebook and Instagram Election Study (Clegg & Nayak, 2020) faced major delays, and initiatives like alternative oversight boards (The Real Facebook Oversight Board, 2022) and coalitions to protect independent researchers (Coalition for Independent Technology Research, 2022) show that the research community and the broader public have not been satisfied with how social media regulation played out. However, we can learn lessons from both past failures and past successes. The encouragement to learn from social media regulation does not mean that we should always take similar regulatory approaches for generative AI. It means that we should carefully assess what worked well, and what needs to be improved, to let these insights inform generative AI regulation.

Third, instead of looking into the past, it may be desirable to think about generative AI regulation from first principles (Clear, 2024). This could prevent getting caught up in unhelpful norms and precedents that may prevent innovative and effective regulation. This is a valid point, but learning from social media regulation and thinking about generative AI regulation from first principles are not mutually exclusive. For example, it is possible to start thinking about desirable regulation free from any other existing ideas, and afterwards analyze whether the developed approaches are promising in light of what we know about technology regulation in other areas. Learning lessons from social media is the best way to prevent avoidable mistakes because many

challenges that generative AI regulation aims to address, including issues such as content moderation and bias, are not as unprecedented as they may seem.

## 6. Conclusion

There are strong disagreements about the approach that should be taken to regulate generative AI. This paper argued that the regulation of generative AI can be informed by the evolution of the regulation of social media. While social media is not the only analogy proposed for generative AI (Maas, 2023), and by no means a perfect analogy, generative AI and social media share key features that make a comparison of the two worthwhile. An analysis of social media regulation efforts—including self-regulation and laws—reveals interesting approaches and best practices. This paper outlined recommendations regarding transparency, researcher access, gathering democratic input, promoting user choice, addressing specific regulatory concerns, increasing investments into computational social science research, and taking on a more global perspective. In the case of social media, self-regulation did not always work, which has resulted in multiple new laws being proposed in the past few years. These self-regulation efforts and laws, including specific approaches to increasing transparency, enhancing user choice, and investing in research, can be valuable pointers for those looking to regulate generative AI. Analyzing social media regulation may inform and accelerate the process of developing generative AI regulation. Regulation takes time and effort, so where possible, we should save resources and avoid mistakes by learning the lessons that social media regulation holds for generative AI regulation.

## Acknowledgements

The author is grateful to Jennifer Pan for feedback on an earlier draft.

## Impact Statement

Against the backdrop of increasingly heated debates about generative AI regulation, this paper shows that we do not have to reinvent the wheel when it comes to questions such as how to ensure that generative AI is safe and moderated in alignment with users' preferences. Instead, we can learn lessons from social media regulation.

Concrete lessons we can learn include the importance of investing in trust and safety and taking a more diverse perspective, both in terms of geography and research disciplines.

Learning lessons from social media regulation can help prevent avoidable mistakes and utilize resources more effectively, which can ultimately improve AI policy and AI safety.

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
