# OpenReview forum: "Position: Generative AI Regulation Can Learn from Social Media Regulation"
_ICML.cc/2025/Position_Paper_Track — ICML 2025 Position Paper Track oral_

### Official Review · Reviewer_DAEG · 2025-03-08

**Significance:** 3
**Argument Clarity:** 2
**Rating:** 4
**Confidence:** 4

**Questions:**

How much level of compliance are social media platforms actually meeting for the various data, transparency, safety regulations? It might be best to have a separate table that lists these required and optional social media regulations and tally which ones are being followed by each platform (e.g., Tiktok, Facebook, X, etc).

**Discussion Potential:**

3

**Paper Summary:**

The position paper argues that regulating generative AI can be informed by looking at existing social media regulations. To support this position, the author/s first highlight comparable and contrasting affordances between social media and generative AI that is split across three categories including medium, control, and immediacy. The author/s emphasize that the alignment between affordances need not be perfect and such similarities in key aspects (e.g., spatial separation, content moderation) can be good source of informing how regulations for AI can be built. The paper then describes for policy recommendations that can be adapted for generative AI including a) countering bias through researcher and user transparency, b) investing in trust and safety, c) promoting CSS research, and d) improving global perspective. Conveniently, the paper summarizes the policy recommendation in Figure 1 reflecting current social media precedents with proposed measures for generative AI.

**Position:**

Yes

**Position In Title:**

Yes

**Related Work:**

3

**Strengths And Weaknesses:**

In terms of strengths, I appreciate that the paper is very well-written, well-organized and readable. This makes conveying the author/s’ position easy for me to understand and to any reader.

In terms of the quality of the position, after reading the paper multiple times, I think the strength of the positiion sits in the middle, somewhat neutral in stance. I’m aware (and the author/s also have said this) that the alignment of affordances between social media and generative AI need not to be perfect. However, this plants an idea to the readers that even if learning from social media regulations for generative AI **can** be informative **but not** a top priority given substantial differences in some aspects. I was also expecting a stronger message framing or sense of urgency from the paper with respect to the same level of urgency required for AI regulations.

One particular recurring problem with regulations, particularly with what we’re heading to for with GenAI regulation such as the EU AI Act, is the risk of widening “regulatory gaps” where stipulations from regulations are too vague to for standard developers to benchmark or evaluate for proper monitoring–leaving SMEs or smaller providers of regulated services (in this case AI model providers) in a limbo (see this whitepaper for more information:  https://carnegieendowment.org/research/2024/03/ai-and-product-safety-standards-under-the-eu-ai-act?lang=en). Thus, I strongly advise the author/s to provide a concrete discussion on how existing social media regulations or practices can be adapted for AI regulation to reduce or alleviate potential risks of regulatory gaps, especially for SMEs or startups that will use AI for their services.

**“To address problems like biases, it is crucial that even small companies take on a global perspective and become global companies with local expertise in multiple countries.”** -> It’s worth mentioning that this is highly unrealistic for starting small companies or startups but already being done by large companies. No small company will immediately have intercontinental branches globally. It might be best to provide a proper scoping of what level of companies (size, popularity, etc) does the recommendations from social media regulation can be sufficiently applied to in case it will be adapted for generative AI regulations. Again, the goal for generative AI regulation is to not stifle innovations from smaller companies/SMEs and to reduce risks of regulatory gaps.

**Support:**

3

---

> ### Author Rebuttal · Authors · 2025-03-31
>
> We are very grateful to Reviewer DAEG for their constructive feedback and appreciation of the clarity of our paper, including our summary figure and the position. The reviewer highlights a number of very valuable points, which we will integrate into the paper.
>
> **However, this plants an idea to the readers that even if learning from social media regulations for generative AI can be informative but not a top priority given substantial differences in some aspects. I was also expecting a stronger message framing or sense of urgency from the paper with respect to the same level of urgency required for AI regulations.**
>
> This is extremely helpful feedback, and we plan to strengthen the motivation and framing to address this. Specially, we believe that AI regulation is urgent, also because we have seen in the shortcomings of social media regulation that timely, thoughtful, and forward-thinking regulation is key. We will make this link and motivation more clear.
>
> Regarding the differences between generative AI and social media, we plan to expand on them in a new Discussion section, and draw on other areas that generative AI could learn from in cases where social media is not a helpful precedent. We will also be clearer in highlighting that for the main focus of the paper–regulation of the content and potential harm of platforms–social media is a helpful and strong point of comparison for generative AI due to its similarities in these areas.
>
> **One particular recurring problem with regulations, particularly with what we’re heading to for with GenAI regulation such as the EU AI Act, is the risk of widening “regulatory gaps” where stipulations from regulations are too vague to for standard developers to benchmark or evaluate for proper monitoring [...] I strongly advise the author/s to provide a concrete discussion on how existing social media regulations or practices can be adapted for AI regulation to reduce or alleviate potential risks of regulatory gaps, especially for SMEs or startups that will use AI for their services.**
>
> We deeply appreciate this point and agree that it is important to discuss regulatory gaps. We plan to do so by 1) mentioning the challenge of regulatory gaps explicitly, also including the citation to the article you mentioned, which we think highlights key points on the need for both clarity and flexibility in legislation, 2) making our recommendations more concrete (e.g., mentioning frameworks like ROOST (https://roost.tools/) suggested by reviewer 2gNX that help implement Trust and Safety frameworks more easily also for small companies), 3) explicitly acknowledging that companies of different sizes may face different challenges that should be addressed by regulation, and 4) adding a discussion of existing social media and AI regulation and how these can inform ways to close the regulatory gap.
>
> **It might be best to provide a proper scoping of what level of companies (size, popularity, etc) does the recommendations from social media regulation can be sufficiently applied to in case it will be adapted for generative AI regulations. Again, the goal for generative AI regulation is to not stifle innovations**
>
> Related to the previous point, we think this is really important to call out more directly, and we will explicitly discuss that companies of different sizes, popularity etc. may require different treatment and may not be able to meet the same requirements. Also in response to reviewer 2gNX, we will add more about existing regulations in this space, which provide templates for scoping (e.g., based on user base).
>
> **How much level of compliance are social media platforms actually meeting for the various data, transparency, safety regulations? It might be best to have a separate table that lists these [...] regulations and tally which ones are being followed by each platform**
>
> This is an excellent question! There is some existing work investigating compliance with both social media regulation (e.g., audits of DSA compliance, legal cases) and even AI regulations, which we will cite. There are certainly instances of noncompliance, and we will discuss this in more detail either in text or as you suggest in a new table.

---

> > ### Comment · Reviewer_DAEG · 2025-04-01
> >
> > This is to acknowledge that I have read the authors' response to my review, the other reviewers' feedback, and the authors' response to those reviews. I thank the authors for the thorough response.
> >
> > The authors seem to provide a clear plan for the next revision of the paper. I trust they will integrate my suggestions, including improving the urgency of the framing of the position, discussion on regulatory gaps, and scope of companies that will benefit from social media-driven AI regulation.
> >
> > I will raise my score from 3 to 4. This paper would be a great addition to the conference.

---

### Official Review · Reviewer_x4bo · 2025-03-13

**Significance:** 3
**Argument Clarity:** 4
**Rating:** 4
**Confidence:** 2

**Questions:**

1. For the characteristics in which GenAI and AI are not the same, could there be some similarities with other industries that could also be exploited?

**Discussion Potential:**

3

**Paper Summary:**

The authors in this papers argue that GenAI regulation can be catalyzed by Social Media Regulation. Moreover, the authors also propose some policy recommendations for GenAI regulation bae on the evolution of social media regulation.
They first compare GenAI and Social Media features, and beside they are note exactly the same, they show that key share key features for regultaions purposes. Then, they start describing the policy recommendations for GenAI based on the evolution of social media regulation: (1) counter bias and perceptions thereof (e.g., via transparency, oversight boards, researcher access, democratic input), (2) address
specific regulatory concerns (e.g., youth wellbeing, election integrity) and invest in trust and safety, (3) promote computational social science research, and (4) take on a more global perspective.

**Position:**

Yes

**Position In Title:**

Yes

**Related Work:**

3

**Strengths And Weaknesses:**

Strengths:
1. Authors position is clear.
2. Authos support using a table of features supports its position clearly.
3. The paper has the potential to inspire useful discussion.
4. Presentation is clear.

Weakness:
1. The abstract is to large.
2. The authors do not deal with features that are not similar to media.

**Support:**

4

---

> ### Author Rebuttal · Authors · 2025-03-31
>
> We thank Reviewer x4bo for their very helpful feedback and are encouraged that they find the position clear, well-presented and think that is has the potential to inspire useful discussion. We are grateful for the constructive feedback, which we plan to integrate into the paper.
>
> **The abstract is to large.**
>
> Thank you for pointing out that the abstract in its current form is a little long. We will work on cutting the word count in the abstract, e.g. leaving out a few examples and streamlining it more.
>
> **The authors do not deal with features that are not similar to media.**
>
> Thank you for encouraging us to discuss features where generative AI differs from social media more. We currently only have a brief discussion of differences in the Alternative Views section. We plan to expand this section into a more detailed discussion, and, to your point below, agree that in those areas, other industries may hold potential lessons to learn from.
>
> **For the characteristics in which GenAI and AI are not the same, could there be some similarities with other industries that could also be exploited?**
>
> This is a really great point! We agree that regulation in other areas beyond social media can also hold important lessons for generative AI regulation. We plan to address this in a new Discussion section that expands on the differences between generative AI and social media, and draws on other comparisons that may be helpful in this space. For example, generative AI and social media differ in that only generative AI chatbots tend to be a direct conversation partner for humans, in contrast to traditional social media platforms where people post to interact with other people. While this distinction is starting to blur with offerings such as Meta AI integrated into Facebook, regulation of other technologies with a direct user connection (e.g., traditional chatbots that are not based on generative AI) could be insightful.

---

### Official Review · Reviewer_LwMj · 2025-03-14

**Significance:** 4
**Argument Clarity:** 4
**Rating:** 5
**Confidence:** 3

**Questions:**

1. Do you think there are significant enough differences in the pace of change of generative AI relative to social media to warrant making parts of the policy recommendations more flexible? It seems like there is always a new version of ChatGPT or Gemini dropping at a rate that is much more significant than, say, major overhauls to a timeline on Facebook or X.

2. Does the difference in regulatory approaches in different parts of the world mean that certain of your recommendations will be more applicable in certain geographic regions?

**Discussion Potential:**

4

**Paper Summary:**

This paper puts forth the position that regulation of generative AI can take lessons from similar debates that occurred regarding regulation of social media. It first discusses the similarities and differences between generative AI systems and social media. It then puts forth a series of policy recommendations for generative AI regulation that draw from similar approaches in social media: content moderation and ways of countering perceptions of bias; focuses on groups that might be particularly vulnerable to harm, such as youth or voters; and taking a more global perspective.

**Position:**

Yes

**Position In Title:**

Yes

**Related Work:**

4

**Strengths And Weaknesses:**

The strengths of this paper are in its comprehensive comparisons of generative AI and social media. The work does an admirable job of illustrating the (sometimes non-obvious) parallels between the two domains. I particularly appreciated the clear layouts of Table 1 and Figure 1 - both were easy to visually comprehend and made the parallels quite clear. The work also has a strong basis in the literature as evidence to support its claims, and it does discuss alternative views. It is thoroughly cited.

There are not many weaknesses in this paper, but one potential area of improvement would be to have more discussion of how the recommended approaches fit into existing regulatory frameworks like the Digital Services Act or EU AI Act. These are both mentioned at a couple of points in the work, but I do think a more thorough treatment of what those do and don’t do relative to the paper’s recommendations would improve the work, since those are the primary regulatory approaches that are out there at the moment. It might also be worth a brief discussion about how the current political climate in the U.S. and attitudes towards big tech might influence the policy questions at play here. Both of these recommendations are not necessary, the paper is quite good as is, but I think they would speak a bit more to how some of these recommendations might be practically implemented in the real world.

**Support:**

4

---

> ### Author Rebuttal · Authors · 2025-03-31
>
> We would like to thank Reviewer LwMj for their very positive feedback, in particular regarding our clear illustrations, evidence base, and comprehensiveness. We really appreciate the suggestion to expand on the relation to existing AI policy measures and the current political context, which we will integrate into the paper.
>
> **There are not many weaknesses in this paper, but one potential area of improvement would be to have more discussion of how the recommended approaches fit into existing regulatory frameworks like the Digital Services Act or EU AI Act. These are both mentioned at a couple of points in the work, but I do think a more thorough treatment of what those do and don’t do relative to the paper’s recommendations would improve the work, since those are the primary regulatory approaches that are out there at the moment.**
>
> This is a great suggestion! We plan to discuss existing regulatory frameworks in more detail, including how they do or do not cover and address the recommendations in this paper. We will likely add this content in a new section on existing regulations or a new Discussion section which we plan to add. On the EU side, the DSA and the EU AI Act are great examples, and we may also add other examples from other international organizations or individual countries.
>
> **It might also be worth a brief discussion about how the current political climate in the U.S. and attitudes towards big tech might influence the policy questions at play here.**
>
> This is a great point. We agree that policy is partly shaped by current context, and this is important to acknowledge. There is some recent research into people's perceptions of AI companies and public opinion on content moderation in AI, and similar research on social media, which we could cite when we add this point.
>
> **Do you think there are significant enough differences in the pace of change of generative AI relative to social media to warrant making parts of the policy recommendations more flexible? It seems like there is always a new version of ChatGPT or Gemini dropping at a rate that is much more significant than, say, major overhauls to a timeline on Facebook or X.**
>
> This is a really great point. We agree that the pace of development and new versions of generative AI models is faster than of most social media platforms, which suggests, as you allude to, that flexible policy with foresight and adaptability to future changes is even more important. We plan to add this point on a need for flexible policy, especially given the pace of development, in a new Discussion section.
>
> **Does the difference in regulatory approaches in different parts of the world mean that certain of your recommendations will be more applicable in certain geographic regions?**
>
> This is an important question. While most of the recommendations seem applicable very broadly across geographic regions, we agree that different existing or future regulations will make the implementation of the recommendation more or less feasible in different geographies. For example, the EU AI Act seems relatively aligned with some of our recommendations and may make implementation easier in EU countries than in countries that pursue different regulations.

---

### Official Review · Reviewer_2gNX · 2025-03-14

**Significance:** 3
**Argument Clarity:** 3
**Rating:** 4
**Confidence:** 5

**Questions:**

- For the "Invest in trust and safety" section, how do recent efforts such as ROOST (https://roost.tools/) fit in around the types of changes you are proposing?
- If gen AI regulation should learn from social media regulation, how can it learn to better deal with the politicization / minefields in the latter (ex. debates around Section 230)?

**Discussion Potential:**

3

**Paper Summary:**

This paper argues that generative AI regulation should be informed by debates and principles around social media regulation. The paper compares and contrasts the affordances of generative AI and social media, and then discusses specific policy recommendations around AI regulation.

**Position:**

Yes

**Position In Title:**

Yes

**Related Work:**

3

**Strengths And Weaknesses:**

Strengths:
- Comprehensive comparison / contrast between AI and social media regulation
- Insightful ways to tie both of these areas together

Weaknesses:
- "AI regulation can learn from social media regulation" is a fairly broad position to be taking. Would be better if there's a way to qualify this position more -- in what way can it learn? You're more specific in the recommendations section of the paper, but I wonder if you could be more explicit up front when you mention your thesis.

**Support:**

4

---

> ### Author Rebuttal · Authors · 2025-03-31
>
> We are very encouraged by Reviewer 2gNX's positive feedback, in particular the view that this is an insightful and comprehensive comparison between generative AI and social media regulation. We respond to the reviewer's suggestions and questions below, which we will integrate into the paper.
>
> **"AI regulation can learn from social media regulation" is a fairly broad position to be taking. Would be better if there's a way to qualify this position more -- in what way can it learn? You're more specific in the recommendations section of the paper, but I wonder if you could be more explicit up front when you mention your thesis.**
>
> We agree that the title is relatively broad and that the specific recommendations appear only later in the paper. We plan to address this by adding more specific recommendations along with the thesis early on in the paper. E.g., after the paragraph starting with “In this paper, we argue that generative AI regulation can learn from social media regulation”, we suggest to add: “Specific learnings for generative AI regulation based on the evolution of social media regulation include investments in (1) efforts to counter bias and perceptions thereof (e.g., via transparency, researcher access, oversight boards, democratic input), (2) specific areas of regulatory concern (e.g., youth wellbeing, election integrity) and trust and safety, (3) computational social science research, and (4) a more global perspective (see Section 3 for detailed recommendations).”
>
> **For the "Invest in trust and safety" section, how do recent efforts such as ROOST (https://roost.tools/) fit in around the types of changes you are proposing?**
>
> Thank you for pointing to ROOST, this fits very well with the proposed recommendations because it (1) enables access to trust and safety resources, even for smaller companies that may not have familiarity with this space or a lot of resources to dedicate to it, (2) is an example of collaboration via open-source tools, in line with the collaboration across platforms that we mention as a helpful development in the social media trust and safety space. We plan to add a reference to ROOST with the aforementioned reasoning because we see it as a helpful resource to the community that makes our recommendations more concrete.
>
> **If gen AI regulation should learn from social media regulation, how can it learn to better deal with the politicization / minefields in the latter (ex. debates around Section 230)?**
>
> This is a great question, and the argument that generative AI regulation should learn from social media regulation certainly implies learning both from things that went well (to take them as inspiration) and things that went wrong (to avoid them). With regards to politicization, we hope that this can be partially addressed by the recommendations in the sections “Counter bias and perceptions thereof”, “Increase transparency and researcher access” and “Gather democratic input to inform technology”. While polarization is a deeper issue in society and not easy to overcome with a single study or transparency measure, we hope that more transparency and insight into genAI, taking user preferences seriously, and being accountable (e.g., fixing issues discovered by researchers) may help focus the debate on objective progress, not politics. We believe that some of the recent and current litigation around Section 230 as it relates to social media platforms is relevant to AI platforms, too, so carefully studying how precedents evolve and what the courts consider social media platforms’ rights and liabilities could help inform AI platforms’ approaches to content moderation and proactive thinking around these issues.

---

### Decision · Program_Chairs · 2025-04-26

**Decision:**

Accept (oral)

**Comment:**

Much of the discussion of AI regulation starts "from first principles", and this paper makes a compelling argument that the conversation should instead start with regulation around social media. While there are still many open questions in that domain, the paper makes a compelling argument that there are many similarities, and a common framework can help guide the evolution of how to think about AI regulation informed by the norms, laws, and expectations that have grown up around AI.

Reviews were uniformly positive, although there are two salient weaknesses that should be addressed in any revision:

1. Going a little further in connecting to existing regulation regimes for social media and how the lessons learned from them (230/DSA) could inform AI regulation
2. discuss how this regulation favors incumbents: increased regulatory compliance burdens may be impossible for startups or smaller markets to overcome